# High-Quality Self-Supervised Deep Image Denoising

**Samuli Laine**
NVIDIA*

**Tero Karras**
NVIDIA

**Jaakko Lehtinen**
NVIDIA, Aalto University

**Timo Aila**
NVIDIA

## Abstract

We describe a novel method for training high-quality image denoising models based on unorganized collections of corrupted images. The training does not need access to clean reference images, or explicit pairs of corrupted images, and can thus be applied in situations where such data is unacceptably expensive or impossible to acquire. We build on a recent technique that removes the need for reference data by employing networks with a "blind spot" in the receptive field, and significantly improve two key aspects: image quality and training efficiency. Our result quality is on par with state-of-the-art neural network denoisers in the case of i.i.d. additive Gaussian noise, and not far behind with Poisson and impulse noise. We also successfully handle cases where parameters of the noise model are variable and/or unknown in both training and evaluation data.

## 1 Introduction

Denoising, the removal of noise from images, is a major application of deep learning. Several architectures have been proposed for general-purpose image restoration tasks, e.g., U-Nets [23], hierarchical residual networks [20], and residual dense networks [31]. Traditionally, the models are trained in a supervised fashion with corrupted images as inputs and clean images as targets, so that the network learns to remove the corruption.

Lehtinen et al. [17] introduced NOISE2NOISE training, where pairs of corrupted images are used as training data. They observe that when certain statistical conditions are met, a network faced with the impossible task of mapping corrupted images to corrupted images learns, loosely speaking, to output the "average" image. For a large class of image corruptions, the clean image is a simple per-pixel statistic — such as mean, median, or mode — over the stochastic corruption process, and hence the restoration model can be supervised using corrupted data by choosing the appropriate loss function to recover the statistic of interest.

While removing the need for clean training images, NOISE2NOISE training still requires at least two independent realizations of the corruption for each training image. While this eases data collection significantly compared to noisy-clean pairs, large collections of (single) poor images are still much more widespread. This motivates investigation of self-supervised training: how much can we learn from just looking at corrupted data? While foregoing supervision would lead to the expectation of some regression in performance, can we make up for it by making stronger assumptions about the corruption process? In this paper, we show that for several noise models that are i.i.d. between pixels (Gaussian, Poisson, impulse), only minor concessions in denoising performance are necessary. We furthermore show that the parameters of the noise models do not need to be known in advance.

We draw inspiration from the recent NOISE2VOID training technique of Krull et al. [14]. The algorithm needs no image pairs, and uses just individual noisy images as training data, assuming that the corruption is zero-mean and independent between pixels. The method is based on *blind-spot networks* where the receptive field of the network does not include the center pixel. This

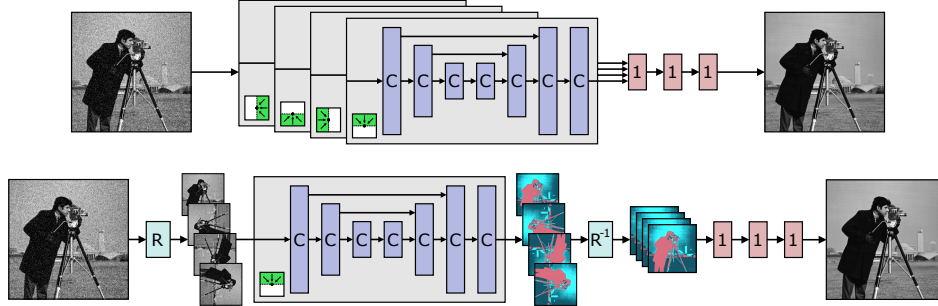

Figure 1: **Top:** In our blind-spot network architecture, we effectively construct four denoiser network branches, each having its receptive field restricted to a different direction. A single-pixel offset at the end of each branch separates the receptive field from the center pixel. The results are then combined by 1×1 convolutions. **Bottom:** In practice, we run four rotated versions of each input image through a single receptive field -restricted branch, yielding a simpler architecture that performs the same function. This also implicitly shares the convolution kernels between the branches and thus avoids the four-fold increase in the number of trainable weights.

allows using the same noisy image as both training input and training target — because the network cannot see the correct answer, using the same image as target is equivalent to using a different noisy realization. This approach is self-supervised in the sense that the surrounding context is used to predict the value of the output pixel without a separate reference image [8].

The networks used by Krull et al. [14] do not have a blind spot by design, but are trained to ignore the center pixel using a masking scheme where only a few output pixels can contribute to the loss function, reducing training efficiency considerably. We remedy this with a novel architecture that allows efficient training without masking. Furthermore, the existence of the blind spot leads to poor denoising quality. We derive a scheme for combining the network output with data in the blind spot, bringing the denoising quality on par with, or at least much closer to, conventionally trained networks.

## 2   Convolutional blind-spot network architectures

Our convolutional blind-spot networks are designed by combining multiple branches that each have their receptive field restricted to a half-plane (Figure 1) that does not contain the center pixel. We combine the four branches with a series of 1×1 convolutions to obtain a receptive field that can extend arbitrarily far in every direction but does not contain the center pixel. The principle of limiting the receptive field has been previously used in PixelCNN [29, 28, 24] image synthesis networks, where only pixels synthesized before the current pixel are allowed in the receptive field.[2] The benefit of our architecture compared to the masking-based training of Krull et al. [14] is that all output pixels can contribute to the loss function as in conventional training.

In order to transform a restoration network into one with a restricted receptive field, we modify each individual layer so that its receptive field is fully contained within one half-plane, including the center row/column. The receptive field of the resulting network includes the center pixel, so we offset the feature maps by one pixel before combining them. Layers that do not extend the receptive field, e.g., concatenation, summation, 1×1 convolution, etc., can be used without modifications.

**Convolution layers**   To restrict the receptive field of a zero-padding convolution layer to extend only, say, upwards, the easiest solution is to offset the feature maps downwards when performing the convolution operation. For an $h \times w$ kernel size, a downwards offset of $k = \lfloor h/2 \rfloor$ pixels is equivalent to using a kernel that is shifted upwards so that all weights below the center row are zero. Specifically, we first append $k$ rows of zeros to the top of input tensor, then perform the convolution, and finally crop out the $k$ bottom rows of the output.

**Downsampling and upsampling layers**   Many image restoration networks involve downsampling and upsampling layers, and by default, these extend the receptive field in all directions. Consider, e.g., a $2 \times 2$ average downsampling step followed immediately by a nearest-neighbor $2 \times 2$ upsampling step. The contents of every $2 \times 2$ pixel block in the output now correspond to the average of this block in the input, i.e., information has been transferred in every direction within the block. We fix this problem by again applying an offset to the data. It is sufficient to restrict the receptive field for the *pair* of downsampling and upsampling layers, which means that only one of the layers needs to be modified, and we have chosen to attach the offsets to the downsampling layers. For a $2 \times 2$ average downsampling layer, we can restrict the receptive field to extend upwards only by padding the input tensor with one row of zeros at top and cropping out the bottom row before performing the actual downsampling operation.

## 3   Self-supervised Bayesian denoising with blind-spot networks

Consider the prediction of the clean value $\boldsymbol{x}$ for a noisy pixel $\boldsymbol{y}$. As the pixels in an image are not independent, all denoising algorithms assume the clean value depends not only on the noisy measurement $\boldsymbol{y}$, but also on the context of neighboring (noisy) pixels that we denote by $\Omega_y$. For our convolutional networks, the context corresponds to the receptive field sans the central pixel. From this point of view, denoising can be thought of as statistical inference on the probability distribution $p(\boldsymbol{x}|\boldsymbol{y}, \Omega_y)$ over the clean pixel value $\boldsymbol{x}$ conditioned with both the context $\Omega_y$ and the measurement $\boldsymbol{y}$. Concretely, a standard supervised regression model trained with corrupted-clean pairs and $L_2$ loss will return an estimate of $\mathbb{E}_{\boldsymbol{x}}[p(\boldsymbol{x}|\boldsymbol{y}, \Omega_y)]$, i.e., the mean over all possible clean pixel values given the noisy pixel and its context.

Assuming the noise is independent between pixels and independent of the context, the blind-spot network introduced by Krull et al. [14] predicts the clean value based purely on the context, using the noisy measurement $\boldsymbol{y}$ as a training target, drawing on the NOISE2NOISE approach [17]. Concretely, their regressor learns to estimate $\mathbb{E}_{\boldsymbol{x}}[p(\boldsymbol{x}|\Omega_y)]$, i.e., the mean of all potential clean values consistent with the context. Batson and Royer [1] present an elegant general formulation for self-supervised models like this. However, methods that ignore the corrupted measurement $\boldsymbol{y}$ at test-time clearly leave useful information unused, potentially leading to reduced performance.

We bring in extra information in the form of an explicit model of the corruption, provided as a likelihood $p(\boldsymbol{y}|\boldsymbol{x})$ of the observation given the clean value, which we assume to be independent of the context and i.i.d. between pixels. This allows us to connect the observed marginal distribution of the noisy training data to the unobserved distribution of clean data:

$$\underbrace{p(\boldsymbol{y}|\Omega_y)}_{\text{Training data}} = \int \underbrace{p(\boldsymbol{y}|\boldsymbol{x})}_{\text{Noise model}} \underbrace{p(\boldsymbol{x}|\Omega_y)}_{\text{Unobserved}} \mathrm{d}\boldsymbol{x} \tag{1}$$

This functional relationship suggests that even though we only observe corrupted training data, the known noise model should help us learn to predict a parametric model for the distribution $p(\boldsymbol{x}|\Omega_y)$. Specifically, we model $p(\boldsymbol{x}|\Omega_y)$ as a multivariate Gaussian $\mathcal{N}(\boldsymbol{\mu}_x, \boldsymbol{\Sigma}_x)$ over color components. For many noise models, the marginal likelihood $p(\boldsymbol{y}|\Omega_y)$ can then be computed in closed form, allowing us to train a neural network to map the context $\Omega_y$ to the mean $\boldsymbol{\mu}_x$ and covariance $\boldsymbol{\Sigma}_x$ by maximizing the likelihood of the data under Equation (1).

The approximate distribution $p(\boldsymbol{x}|\Omega_y)$ allows us to now apply Bayesian reasoning to include information from $\boldsymbol{y}$ at test-time. Specifically, the (unnormalized) posterior probability of the clean value $\boldsymbol{x}$ given observations of both the noisy pixel $\boldsymbol{y}$ and its context is given by Bayes' rule as follows:

$$\underbrace{p(\boldsymbol{x}|\boldsymbol{y}, \Omega_y)}_{\text{Posterior}} \propto \underbrace{p(\boldsymbol{y}|\boldsymbol{x})}_{\text{Noise model}} \underbrace{p(\boldsymbol{x}|\Omega_y)}_{\text{Prior}} \tag{2}$$

From this point of view, the distribution $p(\boldsymbol{x}|\Omega_y)$ takes the role of the prior, encoding our beliefs on the possible $\boldsymbol{x}$s before observing $\boldsymbol{y}$. (Note that even though we represent the prior as a Gaussian, the posterior is generally not Gaussian due to the multiplication with the noise likelihood.) With the posterior at hand, standard Bayesian inference tools become available: for instance, a maximum a posteriori (MAP) estimate would pick the $\boldsymbol{x}$ that maximizes the posterior; we use the posterior mean $\mathbb{E}_{\boldsymbol{x}}[p(\boldsymbol{x}|\boldsymbol{y}, \Omega_y)]$ for all denoising results as it minimizes MSE and consequently maximizes PSNR.

To summarize, our approach consists of (1) standard training phase and (2) two-step testing phase:

(1) Train a neural network to map the context $\Omega_y$ to the mean $\boldsymbol{\mu}_x$ and variance $\boldsymbol{\Sigma}_x$ of a Gaussian approximation to the prior $p(\boldsymbol{x}|\Omega_y)$.

(2) At test time, first feed context $\Omega_y$ to neural network to yield $\boldsymbol{\mu}_x$ and $\boldsymbol{\Sigma}_x$; then compute posterior mean $\mathbb{E}_{\boldsymbol{x}}[p(\boldsymbol{x}|\boldsymbol{y}, \Omega_y)]$ by closed-form analytic integration.

Looping back to the beginning of this section, we note that the estimate found by standard supervised training with the $L_2$ loss is precisely the same posterior mean $\mathbb{E}_{\boldsymbol{x}}[p(\boldsymbol{x}|\boldsymbol{y}, \Omega_y)]$ we seek. Unfortunately, this does not imply that our self-supervised technique would be guaranteed to find the same optimum: we approximate the prior distribution with a Gaussian, whereas standard supervised training corresponds to a Gaussian approximation of the posterior. However, benign noise models, such as additive Gaussian noise or Poisson noise, interact with the prior in a way that the result is almost as good, as demonstrated below.

In concurrent work, Krull at al. [15] describe a similar algorithm for monochromatic data. Instead of an analytical solution, they use a sampling-based method to describe the prior and posterior, and represent an arbitrary noise model as a discretized two-dimensional histogram.

## 4   Practical experiments

In this section, we detail the implementation of our denoising scheme in Gaussian, Poisson, and impulse noise. In all our experiments, we use a modified version of the five-level U-Net [23] architecture used by Lehtinen et al. [17], to which we append three $1\times1$ convolution layers. We construct our convolutional blind-spot networks based on this same architecture. Details regarding network architecture, training, and evaluation are provided in the supplement. Our training data comes from the 50k images in the ILSVRC2012 (Imagenet) validation set, and our test datasets are the commonly used KODAK (24 images), BSD300 validation set (100 images), and SET14 (14 images).

### 4.1   Additive Gaussian noise

Let us now realize the scheme outlined in Section 3 in the context of additive Gaussian noise. We will cover the general case of color images only, but the method simplifies trivially to monochromatic images by replacing all matrices and vectors with scalar values.

The blind-spot network outputs the parameters of a multivariate Gaussian $\mathcal{N}(\boldsymbol{\mu}_x, \boldsymbol{\Sigma}_x) = p(\boldsymbol{x}|\Omega_y)$ representing the distribution of the clean signal. We parameterize the covariance matrix as $\boldsymbol{\Sigma}_x = \mathbf{A}_x^{\mathrm{T}}\mathbf{A}_x$ where $\mathbf{A}_x$ is an upper triangular matrix. This ensures that $\boldsymbol{\Sigma}_x$ is a valid covariance matrix, i.e., symmetric and positive semidefinite. Thus we have a total of nine output components per pixel for RGB images: the three-component mean $\boldsymbol{\mu}_x$ and the six nonzero elements of $\mathbf{A}_x$.

Modeling the corruption process is particularly simple with additive zero-mean Gaussian noise. In this case, Eq. 1 performs a convolution of two mutually independent Gaussians, and the covariance of the result is simply the sum of the constituents [2]. Therefore,

$$\boldsymbol{\mu}_y = \boldsymbol{\mu}_x \quad \text{and} \quad \boldsymbol{\Sigma}_y = \boldsymbol{\Sigma}_x + \sigma^2\mathbf{I}, \tag{3}$$

where $\sigma$ is the standard deviation of the Gaussian noise. We can either assume $\sigma$ to be known for each training and validation image, or we can learn to estimate it during training. For a constant, unknown $\sigma$, we add $\sigma$ as one of the trainable parameters. For variable and unknown $\sigma$, we learn an auxiliary neural network for predicting it during training. The architecture of this auxiliary network is the same as in the baseline networks except that only one scalar per pixel is produced, and the $\sigma$ for the entire image is obtained by taking the mean over the output. It is quite likely that a simpler network would have sufficed for the task, but we did not attempt to optimize its architecture. Note that the $\sigma$ estimation network is not trained with a known noise level as a target, but it learns to predict it as a part of the training process.

To fit $\mathcal{N}(\boldsymbol{\mu}_y, \boldsymbol{\Sigma}_y)$ to the observed noisy training data, we minimize the corresponding negative log-likelihood loss during training [22, 16, 13]:

$$loss(\boldsymbol{y}, \boldsymbol{\mu}_y, \boldsymbol{\Sigma}_y) = -\log f(\boldsymbol{y}; \boldsymbol{\mu}_y, \boldsymbol{\Sigma}_y) = \tfrac{1}{2}[(\boldsymbol{y}-\boldsymbol{\mu}_y)^{\mathrm{T}}\boldsymbol{\Sigma}_y^{-1}(\boldsymbol{y}-\boldsymbol{\mu}_y)] + \tfrac{1}{2}\log|\boldsymbol{\Sigma}_y| + C, \tag{4}$$

where $C$ subsumes additive constant terms that can be discarded, and $f(\boldsymbol{y}; \boldsymbol{\mu}_y, \boldsymbol{\Sigma}_y)$ denotes the probability density of a multivariate Gaussian distribution $\mathcal{N}(\boldsymbol{\mu}_y, \boldsymbol{\Sigma}_y)$ at pixel value $\boldsymbol{y}$. In cases

Table 1: Image quality results for Gaussian noise. Values of $\sigma$ are shown in 8-bit units.

| Noise type | Method | $\sigma$ known? | KODAK | BSD300 | SET14 | Average |
|---|---|---|---|---|---|---|
| Gaussian $\sigma = 25$ | Baseline, N2C | no | 32.46 | 31.08 | 31.26 | 31.60 |
| | Baseline, N2N | no | 32.45 | 31.07 | 31.23 | 31.58 |
| | Our | yes | 32.45 | 31.03 | 31.25 | 31.57 |
| | Our | no | 32.44 | 31.02 | 31.22 | 31.56 |
| | Our ablated, diag. $\boldsymbol{\Sigma}$ | yes | 31.60 | 29.91 | 30.58 | 30.70 |
| | Our ablated, diag. $\boldsymbol{\Sigma}$ | no | 31.55 | 29.87 | 30.53 | 30.65 |
| | Our ablated, $\boldsymbol{\mu}$ only | no | 30.64 | 28.65 | 29.57 | 29.62 |
| | CBM3D | yes | 31.82 | 30.40 | 30.68 | 30.96 |
| | CBM3D | no | 31.81 | 30.40 | 30.66 | 30.96 |
| Gaussian $\sigma \in [5, 50]$ | Baseline, N2C | no | 32.57 | 31.29 | 31.27 | 31.71 |
| | Baseline, N2N | no | 32.57 | 31.29 | 31.26 | 31.70 |
| | Our | yes | 32.47 | 31.19 | 31.21 | 31.62 |
| | Our | no | 32.46 | 31.18 | 31.13 | 31.59 |
| | Our ablated, diag. $\boldsymbol{\Sigma}$ | yes | 31.59 | 30.06 | 30.54 | 30.73 |
| | Our ablated, diag. $\boldsymbol{\Sigma}$ | no | 31.58 | 30.05 | 30.45 | 30.69 |
| | Our ablated, $\boldsymbol{\mu}$ only | no | 30.54 | 28.56 | 29.41 | 29.50 |
| | CBM3D | yes | 31.99 | 30.67 | 30.78 | 31.15 |
| | CBM3D | no | 31.99 | 30.67 | 30.72 | 31.13 |

where $\sigma$ is unknown and needs to be estimated, we add a small regularization term of $-0.1\sigma$ to the loss. This encourages explaining the observed noise as corruption instead of uncertainty about the clean signal. As long as the regularization is gentle enough, the estimated $\sigma$ does not overshoot — if it did, $\boldsymbol{\Sigma}_y = \boldsymbol{\Sigma}_x + \sigma^2\mathbf{I}$ would become too large to fit the observed data in easy-to-denoise regions.

At test time, we compute the mean of the posterior distribution. With additive Gaussian noise the product involves two Gaussians, and because both distributions are functions of $\boldsymbol{x}$, we have

$$p(\boldsymbol{y}|\boldsymbol{x})\,p(\boldsymbol{x}|\Omega_y) = f(\boldsymbol{x};\,\boldsymbol{y},\sigma^2\mathbf{I})\,f(\boldsymbol{x};\,\boldsymbol{\mu}_x,\boldsymbol{\Sigma}_x), \tag{5}$$

where we have exploited the symmetry of Gaussian distribution in the first term to swap $\boldsymbol{x}$ and $\boldsymbol{y}$. A product of two Gaussian functions is an unnormalized Gaussian function, whose mean [2] coincides with the desired posterior mean:

$$\mathbb{E}_{\boldsymbol{x}}[p(\boldsymbol{x}|\boldsymbol{y}, \Omega_y)] = (\boldsymbol{\Sigma}_x^{-1} + \sigma^{-2}\mathbf{I})^{-1}(\boldsymbol{\Sigma}_x^{-1}\boldsymbol{\mu}_x + \sigma^{-2}\boldsymbol{y}). \tag{6}$$

Note that we do not need to evaluate the normalizing constant (marginal likelihood), as scalar multiplication does not change the mean of a Gaussian.

Informally, the formula can be seen to "mix in" some of the observed noisy pixel color $\boldsymbol{y}$ into the estimated mean $\boldsymbol{\mu}_x$. When the network is certain about the clean signal ($\boldsymbol{\Sigma}_x$ is small), the estimated mean $\boldsymbol{\mu}_x$ dominates the result. Conversely, the larger the uncertainty of the clean signal is compared to $\sigma$, the more of the noisy observed signal is included in the result.

**Comparisons and ablations** Table 1 shows the output image quality for the various methods and ablations tested. Example result images are shown in Figure 2. All methods are evaluated using the same corrupted input data, and thus the only sources of randomness are the network initialization and training data shuffling during training. Denoiser networks seem to be fairly robust to these effects, e.g. [17] reports $\pm 0.02$ dB variation in the averaged results. We expect the same bounds to hold for our results as well.

Let us first consider the case where the amount of noise is fixed (top half of the table). The N2C baseline is trained with clean reference images as training targets, and unsurprisingly produces the best results that can be reached with a given network architecture. N2N [17] matches the results.

Our method with a convolutional blind-spot network and posterior mean estimation is virtually as good as the baseline methods. This holds even when the amount of noise is unknown and needs to be estimated as part of the learning process. However, when we ablate our method by forcing the covariance matrix $\boldsymbol{\Sigma}_x$ to be diagonal, the quality of the results suffers considerably. This setup corresponds to treating each color component of the prior as a univariate, independent distribution, and the bad result quality highlights the need to treat the signal as a true multivariate distribution.

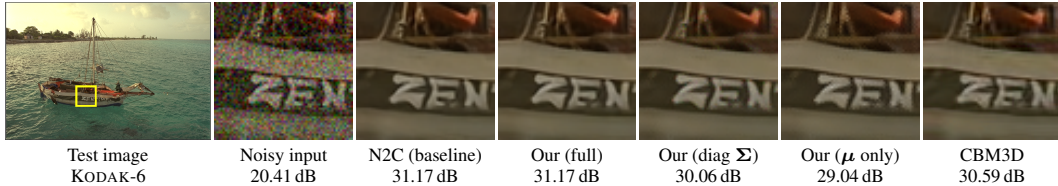

| Test image<br>KODAK-6 | Noisy input<br>20.41 dB | N2C (baseline)<br>31.17 dB | Our (full)<br>31.17 dB | Our (diag $\Sigma$)<br>30.06 dB | Our ($\mu$ only)<br>29.04 dB | CBM3D<br>30.59 dB |

Figure 2: Example result images for methods corresponding to Table 1: Gaussian noise $\sigma = 25$ ($\sigma$ not known). PSNRs refer to the individual images. The supplement gives additional result images, and the full images are included as PNG files in the supplementary material.

Table 2: Average output quality for Gaussian noise ($\sigma = 25$, known) with smaller training sets.

| Method | Training images | | | | | | |
| --- | --- | --- | --- | --- | --- | --- | --- |
| | all | 10 000 | 1000 | 500 | 300 | 200 | 100 |
| Baseline, N2C | 31.60 | 31.59 | 31.53 | 31.44 | 31.35 | 31.21 | 30.84 |
| Our | 31.57 | 31.58 | 31.53 | 31.48 | 31.40 | 31.29 | 31.03 |
| Baseline, N2C + rotation aug. | 31.60 | 31.60 | 31.57 | 31.54 | 31.48 | 31.38 | 31.21 |
| Our            + rotation aug. | 31.58 | 31.58 | 31.53 | 31.47 | 31.42 | 31.32 | 31.08 |

We can ablate the setup even further by having our blind-spot network architecture predict only the mean $\mu$ using standard $L_2$ loss, and using this predicted mean directly as the denoiser output. This corresponds to the setup of Krull et al. [14] in the sense that the center pixel is ignored. As expected, the image quality suffers greatly due to the inability to extract information from the center pixel. Since we do not perform posterior mean estimation in this setup, noise level $\sigma$ does not appear in the calculations and knowing it would be of no use.

Finally, we denoise the same test images using the official implementation of CBM3D [6], a state-of-the-art non-learned image denoising algorithm.[3] It uses no training data and relies on the contents of each individual test image for recovering the clean signal. With both known and automatically estimated (using the method of Chen et al. [5]) noise parameters, CBM3D outperforms our ablated setups but remains far from the quality of our full method and the baseline methods.

The lower half of Table 1 presents the same metrics in the case of variable Gaussian noise, i.e., when the noise parameters are chosen randomly within the specified range for each training and test image. The relative ordering of the methods remains the same as with a fixed amount of noise, although our method concedes 0.1dB relative to the baseline. Knowing the noise level in advance does not change the results.

Table 2 illustrates the relationship between output quality and training set size. Without dataset augmentation, our method performs roughly on par with the baseline and surpasses it for very small datasets ($<$1000 images). For the smaller training sets, rotation augmentation becomes beneficial for the baseline method, whereas for our method it only improves the training of $1\times1$ combination layers. With rotation augmentation enabled, our method therefore loses to the baseline method for very small datasets, although not by much. No other training runs in this paper use augmentation, as it provides no benefit when using the full training set.

**Comparison to masking-based training**   Our "$\mu$ only" ablations illustrate the benefits of Bayesian training and posterior mean estimation compared to ignoring the center pixel as in the original NOISE2VOID method. Here, we shall separately estimate the advantages of having an architectural blind spot instead of masking-based training [14]. We trained several networks with our baseline architecture using masking. As recommended by Krull et al., we chose 64 pixels to be masked in each input crop using stratified sampling. Two masking strategies were evaluated: copying from another pixel in a $5\times5$ neighborhood (denoted COPY) as advocated in [14], and overwriting the pixel with a random color in $[0, 1]^3$ (denoted RANDOM), as done by Batson and Royer [1].

Figure 3: Relative training costs for Gaussian noise ($\sigma = 25$, known) denoisers using the posterior mean estimation. For comparison, training a convolutional blind-spot network for 0.5M minibatches achieves 32.39 dB in KODAK. For the masking-based methods, the horizontal axis takes into account the approximately $4\times$ cheaper training compared to our convolutional blind-spot networks. For example, at $x$-axis position marked "1" they have been trained for 2M minibatches compared to 0.5M minibatches for our method.

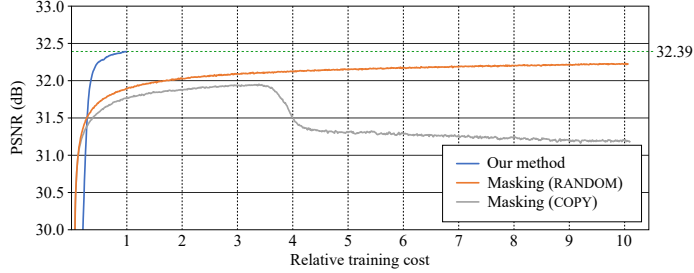

Our tests confirmed that the COPY strategy gave better results when the center pixel was ignored, but the RANDOM strategy gave consistently better results in the Bayesian setting. COPY probably leads to the network learning to leak some of the center pixel value into the output, which may help by sharpening the output a bit even when done in such an *ad hoc* fashion. However, our Bayesian approach assumes that no such information leaking occurs, and therefore does not tolerate it.

Focusing on the highest-quality setup with posterior mean estimation and RANDOM masking strategy, we estimate that training to a quality matching 0.5M minibatches with our convolutional blind-spot architecture would require at least $20$–$100\times$ as much computation due to the loss function sparsity. This is based on a $10\times$ longer masking-based training run still not reaching comparable output quality, see Figure 3.

## 4.2 Poisson noise

In our second experiment we consider Poisson noise which is an interesting practical case as it can be used to model the photon noise in imaging sensors. We denote the maximum event count as $\lambda$ and implement the noise as $y_i = \mathrm{Poisson}(\lambda x_i)/\lambda$ where $i$ is the color channel and $x_i \in [0, 1]$ is the clean color component. For denoising, we follow the common approach of approximating Poisson noise as signal-dependent Gaussian noise [11]. In this setup, the resulting standard deviation is $\sigma_i = \sqrt{x_i/\lambda}$ and the corruption model is thus

$$\boldsymbol{\mu}_y = \boldsymbol{\mu}_x \quad \text{and} \quad \boldsymbol{\Sigma}_y = \boldsymbol{\Sigma}_x + \lambda^{-1}\mathrm{diag}(\boldsymbol{\mu}_x). \tag{7}$$

Note that there is a second approximation in this approach — the marginalization over $\boldsymbol{x}$ (Eq. 1) is treated as a convolution with a fixed Gaussian even though $p(\boldsymbol{y}|\boldsymbol{x})$ should be different for each $\boldsymbol{x}$. In the formula above, we implicitly take this term to be $p(\boldsymbol{y}|\boldsymbol{\mu}_x)$ which is a good approximation in the common case of $\boldsymbol{\Sigma}_x$ being small. Aside from a different corruption model, both training and denoising are equivalent to the Gaussian case (Section 4.1). For cases where the noise parameters are unknown, we treat $\lambda^{-1}$ as the unknown parameter that is either learned directly or estimated via the auxiliary network, depending on whether the amount of noise is fixed or variable, respectively.

**Comparisons** Table 3, top half, shows the image quality results with Poisson noise, and Figure 4, top, shows example result images. Note that even though we internally model the noise as signal-dependent Gaussian noise, we apply true Poisson noise to training and test data. In the case of fixed amount of noise, our method is within 0.1–0.2 dB from the N2C baseline. Curiously, the case where the $\lambda$ is unknown performs slightly better than the case where it is supplied. This is probably a consequence of the approximations discussed above, and the network may be able to fit the observed noisy distribution better when it is free to choose a different ratio between variance and mean.

In the case of variable noise, our method remains roughly as good when the noise parameters are known, but starts to have trouble when they need to be estimated from data. However, it appears that the problems are mainly concentrated to SET14 where there is a 1.2 dB drop whereas the other test sets suffer by only $\sim$0.1 dB. The lone culprit for this drop is the POWERPOINT clip art image, where our method fails to estimate the noise level correctly, suffering a hefty 13dB penalty. Nonetheless, comparing to the "$\boldsymbol{\mu}$ only" ablation with $L_2$ loss, i.e., ignoring the center pixel, shows that our method with posterior mean estimation still produces much higher output quality. Anscombe transform [19] is a classical non-learned baseline for denoising Poisson noise, and for reference we include the results for this method as reported in [17].

Table 3: Image quality results for Poisson and impulse noise.

| Noise type | Method | $\lambda/\alpha$ known? | KODAK | BSD300 | SET14 | Average |
|---|---|---|---|---|---|---|
| Poisson $\lambda = 30$ | Baseline, N2C | no | 31.81 | 30.40 | 30.45 | 30.89 |
| | Baseline, N2N | no | 31.80 | 30.39 | 30.44 | 30.88 |
| | Our | yes | 31.65 | 30.25 | 30.29 | 30.73 |
| | Our | no | 31.70 | 30.28 | 30.35 | 30.78 |
| | Our ablated, $\boldsymbol{\mu}$ only | no | 30.22 | 28.27 | 29.03 | 29.17 |
| | Anscombe [19] (from [17]) | yes | 29.15 | 27.56 | 28.36 | 28.62 |
| Poisson $\lambda \in [5, 50]$ | Baseline, N2C | no | 31.33 | 29.91 | 29.96 | 30.40 |
| | Baseline, N2N | no | 31.32 | 29.90 | 29.96 | 30.39 |
| | Our | yes | 31.16 | 29.75 | 29.82 | 30.24 |
| | Our | no | 31.02 | 29.69 | 28.65 | 29.79 |
| | Our ablated, $\boldsymbol{\mu}$ only | no | 29.88 | 27.95 | 28.67 | 28.84 |
| Impulse $\alpha = 0.5$ | Baseline, N2C | no | 33.32 | 31.20 | 31.42 | 31.98 |
| | Baseline, N2N | no | 32.88 | 30.85 | 30.94 | 31.56 |
| | Our | yes | 32.98 | 30.78 | 31.06 | 31.61 |
| | Our | no | 32.93 | 30.71 | 31.09 | 31.57 |
| | Our ablated, $\boldsymbol{\mu}$ only | no | 30.82 | 28.52 | 29.05 | 29.46 |
| Impulse $\alpha \in [0, 1]$ | Baseline, N2C | no | 31.69 | 30.27 | 29.77 | 30.58 |
| | Baseline, N2N | no | 31.53 | 30.11 | 29.51 | 30.38 |
| | Our | yes | 31.36 | 30.00 | 29.47 | 30.28 |
| | Our | no | 31.40 | 29.98 | 29.51 | 30.29 |
| | Our ablated, $\boldsymbol{\mu}$ only | no | 27.16 | 25.55 | 25.56 | 26.09 |

## 4.3 Impulse noise

Our last example involves impulse noise where each pixel is, with probability $\alpha$, replaced by an uniformly sampled random color in $[0, 1]^3$. This corruption process is more complex than in the previous cases, as both mean and covariance are modified, and there is a Dirac peak at the clean color value. To derive the training loss, we again approximate $p(\boldsymbol{y}|\Omega_y)$ with a Gaussian, and match its first and second raw moments to the data during training. Because the marginal likelihood is a mixture distribution, its raw moments are obtained by linearly interpolating, with parameter $\alpha$, between the raw moments of $p(\boldsymbol{x}|\Omega_y)$ and the raw moments of the uniform random distribution. The resulting mean and covariance are

$$\boldsymbol{\mu}_y = \frac{\alpha}{2} \begin{bmatrix} 1 \\ 1 \\ 1 \end{bmatrix} + (1 - \alpha)\boldsymbol{\mu}_x \quad \text{and} \quad \boldsymbol{\Sigma}_y = \frac{\alpha}{12} \begin{bmatrix} 4 & 3 & 3 \\ 3 & 4 & 3 \\ 3 & 3 & 4 \end{bmatrix} + (1 - \alpha)(\boldsymbol{\Sigma}_x + \boldsymbol{\mu}_x\boldsymbol{\mu}_x^{\mathrm{T}}) - \boldsymbol{\mu}_y\boldsymbol{\mu}_y^{\mathrm{T}}. \quad (8)$$

This defines the approximate $p(\boldsymbol{y}|\Omega_y)$ needed for training the denoiser network. As with previous noise types, in setups where parameter $\alpha$ is unknown, we add it as a learned parameter or estimate it via a simultaneously trained auxiliary network. The unnormalized posterior is

$$\begin{aligned} p(\boldsymbol{y}|\boldsymbol{x})\, p(\boldsymbol{x}|\Omega_y) &= \left(\alpha + (1 - \alpha)\delta(\boldsymbol{y} - \boldsymbol{x})\right) f(\boldsymbol{x}; \boldsymbol{\mu}_x, \boldsymbol{\Sigma}_x) \\ &= \alpha f(\boldsymbol{x}; \boldsymbol{\mu}_x, \boldsymbol{\Sigma}_x) + (1 - \alpha)\delta(\boldsymbol{y} - \boldsymbol{x})f(\boldsymbol{x}; \boldsymbol{\mu}_x, \boldsymbol{\Sigma}_x) \end{aligned} \quad (9)$$

from which we obtain the posterior mean:

$$\mathbb{E}_{\boldsymbol{x}}[p(\boldsymbol{x}|\boldsymbol{y}, \Omega_y)] = \frac{\alpha\boldsymbol{\mu}_x + (1 - \alpha)f(\boldsymbol{y}; \boldsymbol{\mu}_x, \boldsymbol{\Sigma}_x)\boldsymbol{y}}{\alpha + (1 - \alpha)f(\boldsymbol{y}; \boldsymbol{\mu}_x, \boldsymbol{\Sigma}_x)}. \quad (10)$$

Looking at the formula, we can see that the result is a linear interpolation between the mean $\boldsymbol{\mu}_x$ predicted by the network and the potentially corrupted observed pixel value $\boldsymbol{y}$. Informally, we can reason that the less likely the observed value $\boldsymbol{y}$ is to be drawn from the predicted distribution $\mathcal{N}(\boldsymbol{\mu}_x, \boldsymbol{\Sigma}_x)$, the more likely it is to be corrupted, and therefore its weight is low compared to the predicted mean $\boldsymbol{\mu}_x$. On the other hand, when the observed pixel value is consistent with the network prediction, it is weighted more heavily in the output color.

**Comparisons** Table 3, bottom half, shows the image quality results, and example result images are shown in Figure 4, bottom. The N2N baseline has more trouble with impulse noise than with

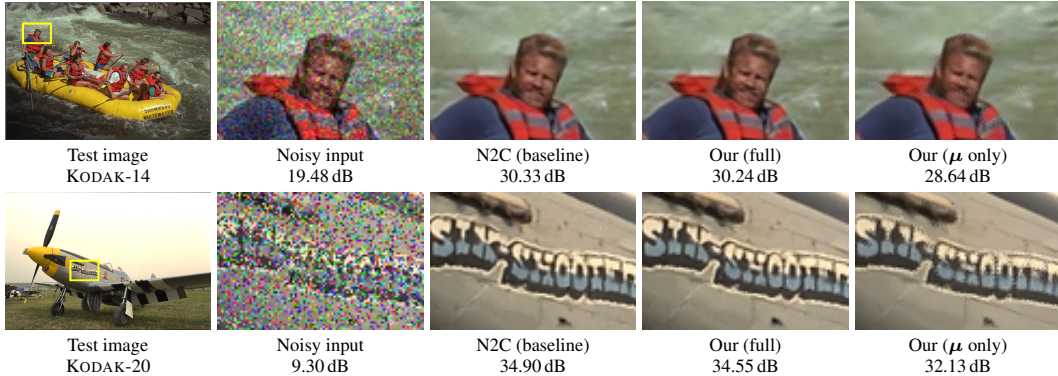

| | | | | |
|---|---|---|---|---|
| Test image KODAK-14 | Noisy input 19.48 dB | N2C (baseline) 30.33 dB | Our (full) 30.24 dB | Our ($\mu$ only) 28.64 dB |
| Test image KODAK-20 | Noisy input 9.30 dB | N2C (baseline) 34.90 dB | Our (full) 34.55 dB | Our ($\mu$ only) 32.13 dB |

Figure 4: Example result images for Poisson (top) and Impulse noise (bottom). PSNRs refer to the individual images. The supplement gives additional result images, and the full images are included as PNG files in the supplementary material.

Gaussian or Poisson noise — note that it cannot be trained with standard $L_2$ loss because the noise is not zero-mean. Lehtinen et al. [17] recommend annealing from $L_2$ loss to $L_0$ loss in these cases. We experimented with several loss function schedules for N2N, and obtained the best results by annealing the loss exponent from 2 to 0.5 during the first 75% of training and holding it there for the remaining training time. Our method loses to the N2C baseline by $\sim$0.4 dB in the case of fixed noise, and by $\sim$0.3 dB with the more difficult variable noise. Notably, our method does not suffer from not knowing the noise parameter $\alpha$ in either case. The ablated "$\mu$ only" setups were trained with the same loss schedules as the corresponding N2N baselines and lose to the other methods by multiple dB, highlighting the usefulness of the information in the center pixel in this type of noise.

## 5    Discussion and future work

Applying Bayesian statistics to denoising has a long history. Non-local means [3], BM3D [7], and WNNM [9] identify a group of similar pixel neighborhoods and estimate the center pixel's color from those. Deep image prior [27] seeks a representation for the input image that is easiest to model with a convolutional network, often encountering a reasonable noise-free representation along the way. As with self-supervised training, these methods need only the noisy images, but while the explicit block-based methods determine a small number of neighborhoods from the input image alone, a deep denoising model may implicitly identify and regress an arbitrarily large number of neighborhoods from a collection of noisy training data.

Stein's unbiased risk estimator has been used for training deep denoisers for Gaussian noise [26, 21], but compared to our work these methods leave a larger quality gap compared to supervised training. Jena [12] corrupts noisy training data further, and trains a network to reduce the amount of noise to the original level. This network can then iteratively restore images with the original amount of noise. Unfortunately, no comparisons against supervised training are given. Finally, FC-AIDE [4] features an interesting combination of supervised and unsupervised training, where a traditionally trained denoiser network is fine-tuned in an unsupervised fashion for each test image individually.

We have shown, for the first time, that deep denoising models trained in a self-supervised fashion can reach similar quality as comparable models trained using clean reference data, as long as the drawbacks imposed by self-supervision are appropriately remedied. Our method assumes pixel-wise independent noise with a known analytic likelihood model, although we have demonstrated that individual parameters of the corruption model can also be successfully deducted from the noisy data. Real corrupted images rarely follow theoretical models exactly [10, 18, 25], and an important avenue for future work will be to learn as much of the noise model from the data as possible. By basing the learning exclusively on the dataset of interest, we should also be able to alleviate the concern that the training data (e.g., natural images) deviates from the intended use (e.g., medical images). Experiments with such real life data will be valuable next steps.

**Acknowledgements**    We thank Arno Solin and Samuel Kaski for helpful comments, and Janne Hellsten and Tero Kuosmanen for the compute infrastructure.

## Footnotes

*{slaine, tkarras, jlehtinen, taila}@nvidia.com

[2]Regrettably the term "blind spot" has a slightly different meaning in PixelCNN literature: van den Oord et al. [28] use it to denote valid input pixels that the network in question fails to see due to poor design, whereas we follow the naming convention of Krull et al. [14] so that a blind spot is always intentional.

[3]Even though (grayscale) WNNM [9] has been shown to be superior to (grayscale) BM3D [7], our experiments with the official implementation of MCWNNM [30], a multi-channel version of WNNM, indicated that CBM3D performs better on our test data where all color channels have the same amount of noise.

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
