[Supplementary Material]

# Supplement: High-Quality Self-Supervised Deep Image Denoising

**Samuli Laine**
NVIDIA

**Tero Karras**
NVIDIA

**Jaakko Lehtinen**
NVIDIA, Aalto University

**Timo Aila**
NVIDIA

## A   Network architecture, training and evaluation details

Table 1 shows the network architecture used in our blind-spot and baseline networks. This is a slightly modified version of the five-level U-Net [9] architecture that was used by Lehtinen et al. [7]. We add three $1{\times}1$ convolution layers at the end in all networks, so that the network depth is the same in both blind-spot and baseline networks. All convolution layers use leaky ReLU [8] with $\alpha = 0.1$, except the very last $1{\times}1$ convolution that has linear activation function.

When forming a blind-spot network, we add three additional layers, denoted ROTATE, SHIFT, and UNROTATE in the table. Layer ROTATE forms four rotated versions (by $0°$, $90°$, $180°$, $270°$) of the input tensor and stacks them on the minibatch axis. Layer SHIFT pads and shifts every feature map downwards by one pixel, thereby raising the receptive field of every pixel upwards by one pixel. This is needed so that when the receptive fields are later combined, the combination excludes the pixel itself. Finally, layer UNROTATE splits the minibatch axis into four pieces, undoes the rotation done in layer ROTATE, and stacks the results on the channel axis, restoring the minibatch size to the original but quadrupling the feature map count. In addition, in blind-spot networks we modify the convolution layers and downsampling layers to extend their receptive field upwards only, as explained in Section 2 of the paper.

**Training and evaluation**   All networks were initialized following He et al. [3] and trained using Adam with default parameters [5], initial learning rate $\lambda = 0.0003$, and minibatch size of 4. The minibatches were composed of random $256{\times}256$ crops from the training set. All networks except those used in impulse noise experiments were trained for 0.5M minibatches, i.e., until 2M training image crops were shown to the network. For the impulse noise experiments we trained the blind-spot networks $2\times$ as long and the baseline networks $8\times$ as long in order to reach convergence. In all training runs, learning rate was ramped down during the last 30% of training using a cosine schedule.

Internally, we use dynamic range of $[0, 1]$ for the image data. The training data was selected to contain only images whose size was between $256{\times}256$ and $512{\times}512$ pixels, in order to exclude images that were too small for obtaining a training crop, or unnecessarily large compared to the test images. We thus used 44328 training images out of the 50k images in ILSVRC2012 validation set. To run the test images through our rotation-based architecture, each of them was padded to a square shape using mirror padding, denoised, and cropped back to original size. To obtain reliable average PSNRs, we replicated each test set multiple times so that each clean image was corrupted by multiple different instances of noise and, in cases with variable noise parameters, different amounts of noise. Specifically, we replicated test sets KODAK, BSD300, and SET14, by 10, 3, and 20 times, yielding average dataset PSNRs that correspond to averages over 240, 300, and 280 individual denoised images, respectively. All methods were evaluated with the same corrupted input data.

The training runs were executed on NVIDIA DGX-1 servers using four Tesla V100 GPUs in parallel. A typical training run took ∼4 hours if using the baseline architecture, and ∼14 hours with the blind-spot architecture due to the fourfold increase in minibatch size inside the network. While training we (unnecessarily) computed the mean posterior estimate for every training crop to monitor convergence, performed frequent test set evaluations, etc., which leaves room for optimizing the training speed.

Table 1: Network architecture used in our experiments. Layers marked with $*$ are present only in the blind-spot variants. Layer NIN_A has 384 output feature maps in the blind-spot networks and 96 in the baseline networks.

| | NAME | $N_{out}$ | FUNCTION |
|---|---|---|---|
| | INPUT | 3 | |
| $*$ | ROTATE | 3 | Rotate and stack |
| | ENC_CONV0 | 48 | Convolution $3 \times 3$ |
| | ENC_CONV1 | 48 | Convolution $3 \times 3$ |
| | POOL1 | 48 | Maxpool $2 \times 2$ |
| | ENC_CONV2 | 48 | Convolution $3 \times 3$ |
| | POOL2 | 48 | Maxpool $2 \times 2$ |
| | ENC_CONV3 | 48 | Convolution $3 \times 3$ |
| | POOL3 | 48 | Maxpool $2 \times 2$ |
| | ENC_CONV4 | 48 | Convolution $3 \times 3$ |
| | POOL4 | 48 | Maxpool $2 \times 2$ |
| | ENC_CONV5 | 48 | Convolution $3 \times 3$ |
| | POOL5 | 48 | Maxpool $2 \times 2$ |
| | ENC_CONV6 | 48 | Convolution $3 \times 3$ |
| | UPSAMPLE5 | 48 | Upsample $2 \times 2$ |
| | CONCAT5 | 96 | Concatenate output of POOL4 |
| | DEC_CONV5A | 96 | Convolution $3 \times 3$ |
| | DEC_CONV5B | 96 | Convolution $3 \times 3$ |
| | UPSAMPLE4 | 96 | Upsample $2 \times 2$ |
| | CONCAT4 | 144 | Concatenate output of POOL3 |
| | DEC_CONV4A | 96 | Convolution $3 \times 3$ |
| | DEC_CONV4B | 96 | Convolution $3 \times 3$ |
| | UPSAMPLE3 | 96 | Upsample $2 \times 2$ |
| | CONCAT3 | 144 | Concatenate output of POOL2 |
| | DEC_CONV3A | 96 | Convolution $3 \times 3$ |
| | DEC_CONV3B | 96 | Convolution $3 \times 3$ |
| | UPSAMPLE2 | 96 | Upsample $2 \times 2$ |
| | CONCAT2 | 144 | Concatenate output of POOL1 |
| | DEC_CONV2A | 96 | Convolution $3 \times 3$ |
| | DEC_CONV2B | 96 | Convolution $3 \times 3$ |
| | UPSAMPLE1 | 96 | Upsample $2 \times 2$ |
| | CONCAT1 | 99 | Concatenate INPUT |
| | DEC_CONV1A | 96 | Convolution $3 \times 3$ |
| | DEC_CONV1B | 96 | Convolution $3 \times 3$ |
| $*$ | SHIFT | 96 | Shift down by one pixel |
| $*$ | UNROTATE | 384 | Unstack, rotate, combine |
| | NIN_A | 384/96 | Convolution $1 \times 1$ |
| | NIN_B | 96 | Convolution $1 \times 1$ |
| | NIN_C | 9 | Convolution $1 \times 1$, linear act. |

**Masking-based training**    In our training runs with masking-based training (end of Section 4.1), we examine convergence by maintaining a smoothed network whose weights follow the trained network using an exponential moving average. This is a commonly used technique in semi-supervised learning (e.g., [10, 1]) and in evaluating Generative Adversarial Networks (e.g., [2, 4]), and removes the need for a learning rate rampdown — and thus deciding the training length in advance — to measure the results near a local minimum.

All curves in Figure 3 were generated by evaluating the test set using this exponentially smoothed network. We verified in separate tests that the results obtained this way were in line with the usual fixed-length training runs with learning rate rampdown.

## B  Additional result images

Figures 1, 2 and 3 show additional denoising results for Gaussian, Poisson, and impulse noise, respectively. In these examples the noise model parameters were fixed but unknown for all algorithms. All PSNRs refer to individual images. We recommend zooming in to the images on a computer screen to better view the differences. The full images are also included as PNG files in the supplementary material.

In this larger set of images we can discern some characteristic failure modes of our ablated setups. When the signal covariance $\Sigma_x$ is forced to be diagonal ("Our ablated, diag. $\Sigma$"), we can see color artifacts on, e.g., rows 6 and 9 of Figure 1. The diagonal covariance matrix corresponds to having a univariate, independent distribution for each color channel, and therefore the network cannot express being, e.g., certain of hue but uncertain of luminance. This may let the color of noise leak through to the result, as seen in some of the images. With full $\Sigma_x$ no such color leaking occurs. The ablation which discards information in center pixel entirely ("Our ablated, $\mu$ only") produces strong pixel-scale diamond/checkerboard artifacts, some of which can also be seen in the results of Krull et al. [6]. In images produced by our full, non-ablated method ("Our"), some slight checkerboarding may be seen in high-frequency areas, especially with impulse noise (see, e.g., Figure 3, bottom row). However, in most cases our results are visually indistinguishable from the baseline results.

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

| | | | | | |
|---|---|---|---|---|---|
| KODAK-6 | 20.41 dB | 31.17 dB | 31.17 dB | 30.06 dB | 29.04 dB | 30.59 dB |
| KODAK-14 | 20.42 dB | 30.88 dB | 30.80 dB | 29.89 dB | 28.98 dB | 30.01 dB |
| KODAK-4 | 20.35 dB | 33.14 dB | 33.09 dB | 32.47 dB | 31.86 dB | 32.66 dB |
| BSD300-18 | 20.30 dB | 31.98 dB | 31.93 dB | 30.77 dB | 29.43 dB | 31.30 dB |
| BSD300-22 | 20.58 dB | 28.63 dB | 28.57 dB | 26.95 dB | 25.07 dB | 28.00 dB |
| BSD300-28 | 20.24 dB | 26.78 dB | 26.78 dB | 24.16 dB | 21.52 dB | 26.37 dB |
| BSD300-80 | 20.20 dB | 32.75 dB | 32.64 dB | 31.81 dB | 31.05 dB | 32.03 dB |
| SET14-2 | 20.36 dB | 31.60 dB | 31.59 dB | 31.30 dB | 30.69 dB | 32.06 dB |
| SET14-5 | 20.51 dB | 29.44 dB | 29.40 dB | 28.34 dB | 26.92 dB | 28.09 dB |
| Test image | Noisy input | N2C baseline | Our | Our ablated, diag. $\Sigma$ | Our ablated, $\mu$ only | CBM3D |

Figure 1: Additional result images for Gaussian noise, $\sigma = 25$.

|  | KODAK-23 | 19.13 dB | 34.83 dB | 34.63 dB | 33.98 dB |
|  | KODAK-8 | 18.63 dB | 29.12 dB | 29.11 dB | 27.25 dB |
|  | BSD300-3 | 21.87 dB | 28.57 dB | 28.43 dB | 23.91 dB |
|  | BSD300-7 | 17.81 dB | 29.39 dB | 29.26 dB | 27.01 dB |
|  | BSD300-11 | 19.91 dB | 31.72 dB | 31.63 dB | 30.01 dB |
|  | BSD300-60 | 18.10 dB | 29.62 dB | 29.61 dB | 27.43 dB |
|  | BSD300-19 | 18.74 dB | 29.49 dB | 29.37 dB | 27.46 dB |
|  | BSD300-21 | 19.33 dB | 31.97 dB | 31.81 dB | 30.24 dB |
|  | BSD300-25 | 19.23 dB | 26.75 dB | 26.65 dB | 23.31 dB |
| Test image | Noisy input | N2C baseline | Our | Our ablated, $\boldsymbol{\mu}$ only |

Figure 2: Additional result images for Poisson noise, $\lambda = 30$.

|   | | | | |
|---|---|---|---|---|
| Kodak-15 | 10.11 dB | 35.13 dB | 34.76 dB | 32.84 dB |
| BSD300-4 | 11.13 dB | 34.41 dB | 33.72 dB | 31.86 dB |
| BSD300-26 | 12.08 dB | 29.39 dB | 28.88 dB | 26.45 dB |
| BSD300-51 | 10.96 dB | 31.36 dB | 30.74 dB | 28.49 dB |
| BSD300-56 | 11.07 dB | 33.03 dB | 32.64 dB | 30.71 dB |
| BSD300-95 | 10.31 dB | 32.41 dB | 31.92 dB | 29.61 dB |
| Set14-1 | 11.19 dB | 23.96 dB | 23.98 dB | 21.81 dB |
| Kodak-20 | 9.30 dB | 34.90 dB | 34.55 dB | 32.13 dB |
| Kodak-19 | 12.09 dB | 33.62 dB | 33.35 dB | 31.08 dB |
| Test image | Noisy input | N2C baseline | Our | Our ablated, $\mu$ only |

Figure 3: Additional result images for impulse noise, $\alpha = 0.5$.