[Reviews · NeurIPS 2019]

Reviewer 1



Originality: Good Quality: Technically sound Clarity: Could use improvement Significance: Moderately significant Weaknesses: -Trains with lots of images (50k); this seems to preclude the methods use in the fields, e.g. biomedical, where you'd like to train without ground truth data. -Only tests synthetic data. -Missing some related work that uses unbiased risk estimators to train denoisers without GT data. These methods already "reach similar quality as compareable models trained using reference clean data". [A] Soltanayev, Shakarim, and Se Young Chun. "Training deep learning based denoisers without ground truth data." Advances in Neural Information Processing Systems. 2018. [B] Metzler, Christopher A., et al. "Unsupervised Learning with Stein's Unbiased Risk Estimator." arXiv preprint arXiv:1805.10531 (2018). [C] Cha, Sungmin, and Taesup Moon. "Fully convolutional pixel adaptive image denoiser." arXiv preprint arXiv:1807.07569 (2018). Minor: On first reading the text, it was unclear that the mean and covariance matrix were across the three color channels. This could be stated explicitly for clarity.

Reviewer 2



1. In the comparison, which method is used as the baseline? Which one is N2C? 2. Why not compare with Noise2Void method?

Reviewer 3



Pros: -The Bayesian analysis with different noise models is interesting. The ablation study is carefully done and confirms the importance of the central pixel integration at test time. This is an important result and may be used in future works. I also find interesting that performance is not too degraded when noise level is unknown. -The experimental results show that their method perform almost as well as their Noise2Clean and Noise2Noise baselines over 3 datasets with different types of noise. It suggests the potential for image denoising using only single instances of corrupted images as training data. -The new convolutional architecture with receptive fields restricted to a half plane is also a nice contribution. The four rotated branches with shared kernels between the branches followed by 1X1 convolutions makes sense. Limitations: -The authors compare their model to their own baselines (Noise2Clean, Noise2Noise). They only provide a comparison with BM3D which is training free. Whereas their method and their baselines are trained on a subset of the imagenet validation set. I think it is important to compare their results to some existing N2C state-of-the-art methods. -In addition, previous N2C models tend to be trained on much smaller datasets. Could the author comment on that ? Does their method give strong performance when trained on a very large dataset only ? Would it be possible to compare their method and their baselines to some existing N2C methods on smaller datasets? -Finally it is not clear to me why the comparison to the masking-based strategy is defer to the supplements because the new architecture is a major contribution of the paper. They claim that their model is 10-100X faster to train than Noise2Void however they are no quantitative experiments in the main text to demonstrate that their architecture is superior to Noise2Void. Looking at section B in the supplement, it seems that even after convergence the masking strategy gives lower performance when evaluated on the kodak dataset (with similar posterior mean estimation). Does their method provide better results than the masking strategy ? Could the author explain why this is the case ?

[Author Response · NeurIPS 2019]

We would like to thank the reviewers for their comments and remarks. We will gladly follow the suggestions for clarifying the paper.

Reviewers #1 and #4 inquired about the quality of our method with smaller training sets. The table below shows the average output quality in dB with various training set sizes on Gaussian noise ($\sigma = 25$). In these tests, our method always performs roughly on par with the baseline supervised training, and with very small training sets appears to consistently outperform it. This may suggest that our method is less prone to overfitting, but investigating the effect further would require additional study. CBM3D yields 30.96 dB in this setup.

| Method | Training images | | | | | | |
| --- | --- | --- | --- | --- | --- | --- | --- |
| | all | 10 000 | 1000 | 500 | 300 | 200 | 100 (10 runs) |
| Baseline, N2C | 31.60 | 31.59 | 31.53 | 31.44 | 31.35 | 31.21 | 30.86 ($\pm$ 0.02) |
| Our | 31.58 | 31.58 | 31.53 | 31.48 | 31.40 | 31.29 | 31.03 ($\pm$ 0.02) |

Reviewer #1 remarked that our experiments are performed on synthetic data only. This decision was motivated by the need to measure denoising quality reliably against a known ground truth, and to compare our method to previous work that commonly follows the same approach of corrupting clean natural images with controlled amounts of synthetic noise. As the non-learned CBM3D method is also designed for natural images, we feel that our comparisons are fair. We agree that experimenting with real medical data would be an important next step.

We would like to thank Reviewer #1 for bringing the SURE-based unsupervised denoising work to our attention. These papers should certainly be cited in our related work section. However, it is not the case that the referred unsupervised SURE-based methods reach similar quality as training an equivalent network in a supervised fashion. Both [1] and [2] as well as our paper contain results in the case of Gaussian noise, $\sigma = 25$. In [1], the average output quality suffers by 0.23 dB in BSD68 (Table 4 in [1], DnCNN-Sure vs. DnCNN-MSE-GT) and by 0.33 dB in SET12 (Table 3) compared to supervised training. In [2] the gap is 0.32 dB averaged over the six test images. As shown in Table 1 of our paper, our method has just a 0.01 dB quality gap to the baseline in KODAK and SET14, and 0.05 dB in BSD300 in the same conditions. This is a much better result than demonstrated in the SURE-based methods. The FC-AIDE method in [3] is not a fully unsupervised method, as it requires ground truth data for training the base network in a supervised fashion, which is then fine-tuned in an unsupervised fashion using the noisy test image at test time. As such, we do not feel that FC-AIDE would be fair as a comparison method, but it could serve as a high-quality baseline.

Reviewers #2 and #4 asked about comparisons to NOISE2VOID. We shall incorporate more of our comparison between masking-based training and architecturally enforced blind spot into the main text as suggested. It is true that we do not compare against plain NOISE2VOID — as we provide two improvements over it, we measure their effects separately. The $\mu$-only ablations are a proxy for NOISE2VOID in that the center pixel is ignored and posterior mean estimation is not done, while we still employ an architectural blind spot. Correspondingly, the masking experiments in Appendix B test the effectiveness of the architecture. We have run an experiment with masking-based training and without posterior mean, i.e., equivalent to the original NOISE2VOID except that the network architecture was the same as in other experiments. In a setting similar to Appendix B, this test converged to 30.31 dB quality at equivalent training time, thus yielding a gap of $\sim$2.1 dB to our result of 32.39 dB. After submission we have extended the training runs in Appendix B considerably further, and as Reviewer #4 suggests, it indeed appears that masking-based training cannot quite match our architecturally enforced blind spot even in the limit. We hypothesize that the network fails to be truly independent of the center pixel, and some noise propagates to the output image, but analyzing this further would be a topic for future work.

Reviewer #4 called for comparisons against state-of-the-art denoising methods. Like in [1] and [2], our focus is on studying the relative efficacy of different training schemes using a known, well-performing network architecture. Answering a question by Reviewer #2, we thus consider the baseline to be the same network trained in a traditional, supervised fashion using clean images as targets, indicated as "Baseline, N2C" in our tables. Achieving absolute state-of-the-art denoising results would likely require significantly more complex architectures, and the increased computational cost makes experimentation difficult (e.g., the NOISE2NOISE paper quotes the training cost of RED30 to be $10\times$ higher than the U-net architecture). However, it would be interesting to evaluate our method with higher-quality architectures in the future and see, e.g., if our ideas could be combined with FC-AIDE [3].

# References

[1] Soltanayev and Chun. Training deep learning based denoisers without ground truth data. In *Adv. NIPS*. 2018.

[2] Metzler et al. Unsupervised learning with Stein's unbiased risk estimator. *CoRR*, abs/1805.10531, 2018.

[3] Cha and Moon. Fully convolutional pixel adaptive image denoiser. *CoRR*, abs/1807.07569, 2018.


[Meta-Review · NeurIPS 2019]

The paper received positive and borderline reviews. After discussion, the reviewers are convinced by the rebuttal; they agree that the proposed approach is interesting and recommend acceptance. The area chair agrees with their assessment and follows their recommendation.